# Nutritional Status, Dietary Intake, and Adherence to the Mediterranean Diet of Children with Celiac Disease on a Gluten-Free Diet: A Case-Control Prospective Study

**DOI:** 10.3390/nu12010143

**Published:** 2020-01-04

**Authors:** Elena Lionetti, Niki Antonucci, Michele Marinelli, Beatrice Bartolomei, Elisa Franceschini, Simona Gatti, Giulia Naspi Catassi, Anil K. Verma, Chiara Monachesi, Carlo Catassi

**Affiliations:** 1Department of Pediatrics, Marche Polytechnic University, 60123 Ancona, Italy; niki.antonucci@gmail.com (N.A.); michele.marinelli@mail.com (M.M.); beatricebartolomei@yahoo.it (B.B.); elisa.franceschini3@gmail.com (E.F.); simona.gatti@hotmail.it (S.G.); giulia.catassi@gmail.com (G.N.C.); anilkrvermaa@gmail.com (A.K.V.); chiara.monachesi28@gmail.com (C.M.); c.catassi@univpm.it (C.C.); 2The Division of Pediatric Gastroenterology and Nutrition and Center for Celiac Research, Mass General Hospital for Children, Boston, MA 02114, USA

**Keywords:** celiac disease, nutritional adequacy, gluten, nutrient, gluten-free diet

## Abstract

The only effective treatment for celiac disease (CD) is a life-long strict gluten-free diet (GFD). Nutritional adequacy of the GFD has remained controversial and a matter of debate for a long time. No large case-control studies on children regarding the nutritional adequacy of the GFD has been performed. In this study, children diagnosed with CD on a GFD for ≥ 2 years were recruited. Controls were age and gender-matched healthy children not affected with CD. In both groups, anthropometric measurements and energy expenditure information were collected. Dietary assessment was performed by a 3-day food diary. Adherence to the Mediterranean diet was estimated by the KIDMED index. Overall, 120 children with CD and 100 healthy children were enrolled. No differences were found between CD children and controls in anthropometric measurements and energy expenditure. In the CD group, the daily intake of fats was significantly higher while the consumption of fiber was lower in comparison with the control group. The median KIDMED index was 6.5 in CD children and 6.8 in healthy controls. The diet of children with CD in this study was nutritionally less balanced than controls, with a higher intake of fat and a lower intake of fiber, highlighting the need for dietary counseling.

## 1. Introduction

Celiac disease (CD) is a systemic immune-mediated disorder caused in genetically susceptible persons by the ingestion of gluten-containing grains [1]. The only available treatment is the gluten-free diet (GFD), which consists of the dietary exclusion of grains containing gluten (i.e., wheat, rye, barley) [2]. 

The nutritional adequacy of the GFD remained controversial and a matter of debate for a long time [3]. Indeed, apart from maintaining the safe limit of gluten intake (below 10–50 mg/day), a suitable GFD must also be nutritionally balanced and cover all energy and nutrient requirements to prevent deficiencies and ensure a healthy life. In children, the GFD must also allow appropriate growth and pubertal development [4]. A body of evidence has so far suggested that the GFD may be nutritionally unbalanced either because of the need to exclude several cereals or because of the different nutritional composition of GF products as compared to their unrefined analogs [3,5,6,7,8,9,10,11,12,13,14,15,16].

To the best of our knowledge, there are no large case-control studies performed on children regarding the nutritional adequacy of the GFD. Previous studies have mostly been performed on adolescents or adults, with the limit of (1) small sample sizes, (2) lack of a control group, (3) retrospective methods of dietary recording, and (4) inclusion of patients at diagnosis.

Therefore, we aimed to evaluate the nutritional status, the dietary intake and adherence to the national recommended dietary allowances as well as to the Mediterranean diet of Italian children with CD on the GFD by a large, prospective case-control study.

## 2. Material and Methods

### 2.1. Study Population

This is a case-control prospective study conducted at the Center for Celiac Disease of the Polytechnic University of Marche from January 2017 to January 2019. All children (age range = 4–16 years) with a diagnosis of CD according to the ESPGHAN criteria [17], on a GFD for ≥ 2 years, were recruited as the CD-group. Patients who (1) had other chronic conditions (including type 1 diabetes or inflammatory bowel disease) or (2) did not adhere to the GFD (as demonstrated by elevation of serologic CD markers at enrollment) were excluded. Controls were healthy age- and gender-matched children not affected with CD (on the basis of a negative result of the IgA class anti-transglutaminase test), participating in a previously described mass screening program for CD [18]. Children with comorbidities or following a special diet for other reasons (vegetarian, vegan diet, or related to particular religious or social traditions) were excluded.

### 2.2. Anthropometric Measurements

For all children, anthropometric measurements were collected by the same trained operator. Body weight was measured using the same mechanical balance (mod. 200, SECA, Limbiate, Italy); height was measured to the nearest 5 mm using a stadiometer (mod. 220, SECA, Limbiate, Italy). Body mass Index (BMI) was calculated from weight and height (Kg/m^2^). The BMI values were categorized according to the World Health Organization criteria as follows: below 18.5 kg/m^2^ considered as underweight, 18.5–24.9 kg/m^2^ as normal weight, 25–29.9 kg/m^2^ as overweight and >30 kg/m^2^ as obese.

### 2.3. Physical Activity

For all children, information about lifestyle, such as the number of weekly hours devoted to physical activity and number of daily hours devoted to sedentary activities (e.g., sitting down in front of the TV, PC, tablet, PlayStation or board games) were collected by a detailed questionnaire.

### 2.4. Dietary Assessment

In both groups, dietary intake was assessed using a 3-day food diary, two on weekdays and one at the weekend. The diary was carefully explained by the same trained dietitian to both children and their parents and was accompanied by detailed instructions for the compilation and a photographic atlas including different portion-size food pictures and a set of about 60 actual household measures. The diary was specifically developed for CD patients and included a daily record of all foods consumed during the different meals (breakfast, morning snack, lunch, afternoon snack, dinner). For each meal, participants were requested to report an exhaustive description of food and recipes (including cooking and preservation methods, sugar or fats added during meal preparation), food amount (according to the atlas) and brand of packaged foods consumed. 

All diaries were analyzed by the same trained dietician using an Excel spreadsheet (specifically developed for the study) to estimate the composition of the macronutrients of the diet and the frequency of foods. In the database each consumed food was classified into the main food group categories: sugary drinks, meat, processed meat, vegetables, fruit, milk and dairy products, legumes, potatoes, fish, eggs and cereals (including pasta, bread and bakery products, rice, minor cereals—e.g. oats- and pseudo-cereals—e.g. buckwheat and quinoa), sweets and salty snacks. Each food group had several subgroups (i.e., cereals had 5 subgroups: pasta, bread products, pizza, rice, minor cereals and pseudo-cereals), and each subgroup was further classified according to its composition (i.e., bread was divided in whole grain bread, type 0 flour bread, type 00 flour bread, milk bread, durum wheat bread, rye bread, etc.), allowing us to estimate the different nutritional composition of each consumed food. The source of information of the nutritional composition of foods was the Italian Food Composition Database [19]. The composition of GF products was retrieved from product labels. Weight of consumed foods was calculated based on the weight of raw foods, as recommended [19]. By using portion size photos, the weight of the portion for different foods was obtained by the guidelines of the Italian Society of Human nutrition [20]. In the presence of several ingredients, we included in the database only foods reaching the size of a portion.

The program estimated the energy intake (Kilocalories), and macronutrients (proteins, total fats, saturated fast, carbohydrates, simple sugars, and fiber—expressed in grams) and the percentage of energy provided by each macronutrient. “National Recommended Energy and Nutrient Intake Levels” (LARN) issued by the Italian Society of Human Nutrition in 2014 [20] and the “Italian Food Pyramid” (IFP) recommended by the Italian Society of Pediatrics [21] were taken as reference values for energy and nutrient intake and for food group consumption, respectively. For CD patients, the impact of commercial GF products specifically formulated for CD in terms of energy and macronutrients was also estimated. 

The adherence to the Mediterranean diet was estimated by the KIDMED index (Mediterranean Diet Quality Index in Children and Adolescents) [22], widely used as an indicator of healthy dietary habits. This index is determined from a 16-point questionnaire that assesses various dietary habits. Each answer is scored according to whether it is consistent with habits associated with the Mediterranean pattern, and scores are added up to quantify the total index of the subject’s adherence to the Mediterranean diet (MD). The KIDMED index ranges from 4 (no adherence to the MD) to 12 (complete adherence to the MD) [22]. The KIDMED test and scoring is attached as Appendix A. The study protocol was approved by the Institutional Review Board of the Polytechnic University of Marche (Ancona, Italy). All subjects gave their informed consent for inclusion before they participated in the study.

### 2.5. Statistical Analysis 

Based on a previous pediatric study [23] and our preliminary data, considering an expected mean BMI of 16 in healthy children, and a mean BMI of 17 in CD children, with a level of significance of 0.05 and a power of 90%, a minimum sample of 95 CD children was calculated. Subjects’ general characteristics were summarized using descriptive statistics: median, first and third quartiles for quantitative variables, and absolute and percent frequencies for qualitative variables. Comparisons between the two groups were performed by means of the Wilcoxon rank-sum test and Fisher test, respectively. The comparison between the estimated levels of energy, macronutrients and food groups consumption references values (LARN and IFP) was carried out using the 95% Confidence Interval (95% CI) for the median. A probability of 0.05 was chosen to assess the statistical significance; the R program (Institute of Statistics and Mathematics, Vienna, Austria) was used for statistical analysis. 

## 3. Results

### 3.1. Study Population

The net participation rate was 90%. The main reasons for refusal were lack of time, no interest, and difficulty to reach the study center (Figure 1). Overall, 120 CD children were enrolled; there were 72 females (60%), the median age was 10.5 (range: 4.4–15.5 years), with a median duration of GFD of 2.6 years (first and third quartiles, 1.4–4.3 years). The control group included 100 healthy children, 56 females (56%), with a median age of 10.1 (range: 4.7–14.5 years). 

### 3.2. Anthropometric Results and Energy Expenditure 

As shown in Table 1, no differences were found between CD children and the control group as regards anthropometric measurements and energy expenditure. In detail, the median BMI was 16.8 in CD children and 16.0 in the control group, with no significant difference between the groups, and the prevalence of overweight and obesity was similar. The self-reported physical activity, as well as the number of daily hours devoted to sedentary activities, were comparable.

### 3.3. Total Energy, Macronutrient Intakes and Adherence to LARN

Table 2 shows the total daily energy and the macronutrient intakes in the two study groups, and the comparison with the LARN recommendations. The estimate of daily energy intake was similar in the two groups. Protein consumption did not differ between CD and control children, and both the daily protein intake and the daily energy intake provided by proteins were in line with the LARN recommendations in both groups. The daily intake of carbohydrates and the energy intake provided by carbohydrates were significantly lower in the CD group (209.7 g in the CD group versus 260.5 g in the control group; *p* = 0.001), although in both groups the percentage of energy supplied by carbohydrates reached the LARN recommendations. The daily intake of simple sugars and their contribution to the daily energy intake were significantly different between the two groups, with a higher intake in the control group. Furthermore, the daily intake of simple sugars exceeded the LARN recommendations (<15% of total energy) only in the control group.

The daily intake of total fats and saturated fats were significantly higher in the CD group (total fats: 78.1 g in the CD group versus 64.4 g in the control group; *p* = 0.015; saturated fats: 25.3 g in the CD group versus 18.7 g in the control group; *p* = 0.003). Indeed, the energy intake provided by total fats and saturated fats was significantly higher in the CD group and exceeded the nutritional goal recommended by LARN (<10% Total Energy). Finally, the daily consumption of fiber was significantly different in the two groups, with a lower daily intake in the CD group (12.6 g in the CD group versus 15 g in the control group; *p* = 0.015); moreover, the energy intake provided by fibers was lower as compared to the appropriate intake suggested by LARN (at least 1.7%), while in healthy controls it reached the lower normal limit.

### 3.4. Food Group Intake and Adherence to IFP

Table 3 shows daily food group consumption, as collected by the 3-day food diary. CD children showed a higher consumption of processed meat and salty snacks as compared to healthy children (2.5 portions in the CD group vs 2 in the control group; *p* = 0.009, and 1 portion vs. 0; *p* = 0.001, respectively). Both groups did not reach the number of portions recommended by the IFP for legumes, vegetables, eggs, and fish, while exceeding in the consumption of sugary drinks, meat and processed meat. The consumption of cereals, milk and dairy products and potatoes reached the IFP recommendations in both groups; in the group of cereals, the consumption of pseudo-cereals was very low in the CD group, and the major contributors were GF products specifically formulated for CD. The consumption of fruit reached the minimum intake recommended by the IFP.

### 3.5. Impact of Commercial GF Products Specifically Formulated for CD

Commercial gluten-free products specifically formulated for CD contributed to 73% of daily carbohydrates, 59% of fibers, 34% of sugars, 28% of total fats, 25% of saturated fats and 22% of proteins. Finally, commercial gluten-free products specifically formulated for CD provided 46% of the total daily energy.

### 3.6. KIDMED Index

The median KIDMED index was 6.5 in CD children and 6.8 in healthy controls, showing a suboptimal adherence to the Mediterranean diet in both groups. 

## 4. Discussion

The present case-control study shows that the nutritional status of CD children does not differ from healthy children. However, the diet of CD children in this study was nutritionally less balanced than controls, with a higher intake of fat and a lower intake of fiber, highlighting the need for dietary counseling.

Data from the literature on the effects of GFD on anthropometric parameters of patients with CD are controversial. On the one hand, it has been reported that a good compliance to the GFD is associated with a positive effect on anthropometric parameters with a recovery of lean body mass, normalization of BMI in both underweight and overweight children, and acceleration of linear growth [23,24,25,26]. On the other hand, there are also studies suggesting that the GFD may have a negative effect on body composition and anthropometric parameters in CD patients, with an increased prevalence of overweight and obesity [5,27,28]. These conflicting data may in part be caused by differences in the duration of the GFD at the time of anthropometric assessment or by the lack of a control group. Our study is the first to evaluate the BMI in a large sample of CD children on a GFD for at least two years as compared to healthy children, showing that there is no difference in the percentage of underweight, normal weight and overweight/obesity between groups. We also evaluated the energy expenditure in the two study groups through lifestyle analysis, showing no differences between CD children and the control group. The similar BMI in the presence of a similar lifestyle suggested that energy intake was similar in the two study groups. Indeed, we did not observe any difference in total daily energy intake.

Nonetheless, concern about the nutritional quality of the GFD emerges from our results. Indeed, by the analysis of 3-day food diaries, we found a higher intake of fat and a lower intake of fiber and carbohydrates in CD children on a GFD as compared to healthy children, while there was no difference in the daily intake of protein. As regards carbohydrates, when comparing the daily intake of macronutrients of CD and control children with the Italian recommendations, the percentage of energy supplied by carbohydrates was, however, in line with the LARN recommendations in both groups. Noticeably, healthy children exceeded the daily intake of simple sugars as compared to LARN recommendations, while CD children did not. The main concern about GFD was the higher consumption of total and saturated fats observed in CD children, with the intake of saturated fat exceeding the nutritional goal recommended by LARN only in the CD group. The intake of fiber was also a concern, being lower in CD children as compared to controls and to LARN recommendations.

Our findings are in line with several previous studies that compared the intake of macronutrients in CD patients with the national recommendations, showing, overall, that CD patients consume less fiber and more fats than recommended [5,7,9,10,11,13]. When comparing the nutritional quality of CD patients on a GFD to that of healthy controls, previous studies showed conflicting results. Consistent with our findings, several studies in adults reported a higher intake of fats in CD patients as compared to healthy subjects [6,8,13], while others reported a lower intake of carbohydrates and protein [5] or only a lower intake of fiber [7] or no differences in CD adolescents as compared to a control group [12]. Finally, Zuccotti et al. showed a higher intake of carbohydrates and lower consumption of fat in 18 CD children as compared to 18 healthy controls by a 24 h recall [11]. Differences between studies may be explained by the different age of patients studied (children versus adolescents and adults), the small sample size of many previous studies, the different methods of dietary collection (prospective food diary versus retrospective recall), and finally by the inclusion of patients both at diagnosis and on GFD in some of the studies. Our study firstly evaluated prospectively the macronutrient intake in a large sample of CD patients of pediatric age with at least 2 years’ experience of GFD as compared to healthy subjects by 3-day food diary that is one of the best-practice methods to obtain dietary data [29]. 

One of the main factors that could explain the unbalanced intakes of nutrients is the dietary pattern. For this reason, in our study, we compared the dietary habit of CD children and healthy subjects with respect to the IFP, showing that: (a) CD children have a higher consumption of processed meat and salty snacks as compared to healthy children; (b) both groups did not reach the portions recommended by the IFP for legumes, vegetables, eggs and fish, while exceeding the consumption of sugary drinks, meat and processed meat; (c) the consumption of minor and pseudo-cereals was very low in the CD group, and the major contributors to cereals were gluten-free products. These results may explain the higher intake of fat and lower intake of fiber observed in CD children, however, they also highlight that the dietary habits of Italian children, either celiacs or controls, are not fully adherent to the Mediterranean diet. Indeed, the KIDMED index was moderate in both groups.

Furthermore, several studies have shown that the nutritional profile of GF products specifically formulated for CD patients is different with respect to regular foods, with a higher content of fat and saturated fat, salt, sugar and a lower content of fiber [14,15]. In our study, commercial GF products specifically formulated for CD patients provided 46% of the total daily energy, thus playing a major role in influencing the imbalance in the diet of CD children. Many GF foods are prepared from refined maize flour and white rice, which are lower in fiber (2.6 g and 0.7 g per 100 g, respectively) than wheat (3.5 g per 100 g) or whole wheat (9.6 g per 100 g) [19]. The exclusion of gluten and the use of only GF raw materials as ingredients result in GF food which is less palatable than regular foods; consequently, the manufacturing of GF foods requires the addition not only of some additives, such as hydrocolloids, but also of some macronutrients, such as fats in the final products to mitigate the loss of gluten. Our study highlights the need of enhancing the nutritional quality of GF products.

The main weakness of the present study was the lack of data on micronutrient intake, an important piece in the puzzle of the nutritional quality of the GFD. This limitation was related to the lack of tables on GF products indicating the micronutrient content. Therefore, it was not possible to accurately evaluate the corresponding intake in the diet. Second, potential recording errors including inaccurate estimates of portions consumed and omission of foods (either deliberate or unintentional) could result in an underestimation of nutritional intake, as in all food diary recording. Finally, results on food group intake were collected by the 3-day food diary, that is not the best instrument to estimate the consumption frequency of some foods, such as those that are not eaten daily (e.g., legumes, fish, egg).

## 5. Conclusions

Our study, together with a review of the literature, highlights the need for celiac patients to receive dietary counseling, a fundamental tool to teach the patient to increase the consumption of naturally gluten-free products, to reduce processed ones, to increase the intake of cereals such as oats, rice, minor and pseudo-cereals, and to adhere to the rules of the Mediterranean diet. 

Nonetheless, our study underlines the need for an adequate nutritional educational program and healthcare policies also for healthy children to ameliorate nutrient intake during childhood, possibly impacting on long-term health outcome.

## Figures and Tables

**Figure 1 nutrients-12-00143-f001:**
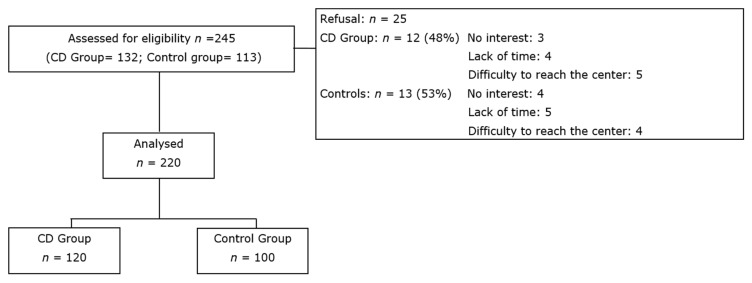
Flow diagram of the study.

**Table 1 nutrients-12-00143-t001:** Anthropometric characteristics and energy expenditure of CD children and control group.

Variable	CD Group (*n* = 120)	Control Group (*n* = 100)	*p*
Age (years)			0.494
Median	10.5	10.1	
First to third quartiles	8.3–12.2	9.4–13.5	
Weight (Kg)			0.448
Median	31.5	32.5	
First to third quartiles	25.2–42.0	28.2–50.0	
Height (cm)			0.26
Median	1.4	1.4	
First to third quartiles	1.2–1.5	1.4–1.5	
Body mass index (Kg/m^2^)			0.988
Median	16.8	16.0	
First to third quartiles	15.3–18.9	15.2–19.5	
Body mass index class (Kg/m^2^)			0.35
Underweight-*n* (%)	6 (5)	5 (5)	
Adequate weight-*n* (%)	78 (65)	70 (70)	
Overweight-*n* (%)	24 (20)	18 (18)	
Obese-*n* (%)	12 (10)	7 (7)	
Physical activity (hours/week)			0.729
Median	4	4	
First to third quartiles	3–8	4–5	
Sedentary activity (hours/week)			0.247
Median	2	2	
First to third quartiles	1–2	2–3	

**Table 2 nutrients-12-00143-t002:** Daily intake of energy, macronutrients of CD children and the control group, and comparison with National Recommended Energy and Nutrient Intake Levels (LARN).

Group		CD Group (*n* = 120)	Control group (*n* = 100)	*p*	LARN
		Median (1st; 3rd Quartiles)	Median (1st; 3rd Quartiles)		
		95% Confidence Interval	95% Confidence Interval		
Energy	Kcal	1819.3 (1589.6; 1997.0)	1838.2 (1782.2; 1964.6)	0.225	
		1715.5–1923.0	1775.3–1901.1		
Total Proteins	grams	59.4 (53.4; 67.7)	65.9 (60.7; 72.0)	0.095	
		55.7–63.0	62.0–69.8		31–62 (PRI)
	% Energy	13.4 (11.8; 14.4)	14.1 (12.4; 14.7)	0.335	
		12.7–14.0	13.4–14.9		10–15% (RI)
	grams/Kg	1.9 (1.5; 2.6)	1.9 (1.7; 2.2)	0.670	
		1.6–2.2	1.7–2.1		0.90–0.99 (PRI)
Total Carbohydrates	grams	209.7 (184.1; 252.1)	260.5 (245.4; 298.4)	**0.001 ***	
		192.5–226.8	242.2–278.8		
	% Energy	46.9 (42.6; 51.7)	53 (50.5; 56.8)	**0.001 ***	
		44.5–49.2	50.9–55.2		45–60% (RI)
Total sugars	grams	68.1 (49.1; 83.3)	83.1 (69.7; 95.3)	**0.036 ***	
		59.4–76.9	74.2–91.9		
	% Energy	14.5 (10.4; 17.6)	17.6 (14.2; 19.6)	**0.036 ***	
		12.6–16.3	15.8–19.4		<15% (SDT)
Total fats	grams	78.1 (63.9; 92.2)	64.4 (59.5; 74.4)	**0.015 ***	
		70.8–85.4	59.2–69.5		
	% Energy	37.5 (32.8; 40.5)	30.5 (28.7; 32.3)	**0.001 ***	
		35.6–39.5	29.3–31.8		20–35% (RI)
Saturated fats	grams	25.3 (20.2; 30.8)	18.7 (16.5; 21.7)	**0.003 ***	
		22.5–28.1	16.9–20.4		
	% Energy	12.8 (10; 14.7)	8.8 (7.8; 10.4)	**0.001 ***	
		11.6–14.0	7.9–9.7		<10% (SDT)
Total Fiber	grams	12.6 (10.9; 16.7)	15 (13.5; 19.1)	**0.015 ***	
		11.1–14.2	13.1–16.9		
	% Energy	1.4 (1.1; 1.7)	1.7 (1.4; 1.9)	0.067	
		1.2–1.5	1.5–1.9		At least 1.7% (AI)

PRI: Population Reference Intake; RI: Reference Intake range for macronutrients; SDT: Suggested Dietary Target; AI: adequate intake. *: Statistically significant

**Table 3 nutrients-12-00143-t003:** Food group consumption in CD children and the control group, and comparison with the recommendations of the Italian Food Pyramid. Consumption is expressed as median of portions consumed over 3 days of recording.

Food Groups	CD Group (*n* = 120)	Control Group (*n* = 100)	*p*	IFP
	Median (1; 3 Quartiles)	Median (1; 3 Quartiles)		
	IC 95%	IC 95%		
Sugary drinks	1 (0; 2)	2 (1; 2)	0.401	
	0.5–1.5	1.5–2.5		Lowest consumption
Meat	2 (1; 3)	2 (1; 3)	0.831	
	1.5–2.5	1.5–2.5		1
Processed Meat	2.5 (2; 4)	2 (1; 2)	**0.009 ***	
	2.0–3.0	1.5–2.5		1
Cheese	1 (0; 3)	1 (0; 2)	0.431	
	0.5–2.0	0.5–1.5		1
Fruits	2.5 (1; 4)	3 (2; 5)	0.121	
	1.5–3.0	2.0–4.0		3–6
Milk and yogurt	3 (1; 3)	3 (3; 4)	0.116	
	2.5–3.5	2.5–3.5		3–6
Legumes	0 (0; 0)	0 (0; 1)	0.175	
	0–0	0–0.5		2
Potatoes	1 (0; 1)	1 (0; 1)	0.475	
	0.5–1.0	0.5–1.5		1
Fish	1 (0; 1)	1 (0; 2)	0.121	
	0.5–1.0	0.5–1.5		1–2
Eggs	0 (0; 1)	0 (0; 1)	0.569	
	0.0–0.5	0.0–0.5		1
Vegetables	2.5 (1; 4)	3 (2;4)	0.122	
	1.5–3.0	2.5–3.5		6–9
Total cereals	15.5 (13; 17)	15 (14; 18.5)	0.463	
	14.5–16.5	13.5–16.5		9–15
Maize	0 (0; 0)	0 (0; 0)	0.906	
	0–0	0–0		Not indicated
Rice	1 (0; 1)	1 (0; 1)	1	
	0.5–1.0	0.5–1.5		Not indicated
Minor and pseudo-cereals	0 (0; 0)	0 (0; 0)	0.167	
	0–0	0–0		Not indicated
Salty snaks	1 (0; 2)	0 (0; 0.5)	**0.001 ***	Lowest consumption
	0.5–1.5	0–0.5		

*: Statistically significant.

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
