# Peer review of "Nutritional Status, Dietary Intake, and Adherence to the Mediterranean Diet of Children with Celiac Disease on a Gluten-Free Diet: A Case-Control Prospective Study"

_nutrients, 2020, doi:10.3390/nu12010143_

Round 1

Reviewer 1 Report

This is an important paper on dietary intake (energy and macronutrients) of children with celiac disease compared to the dietary intake of healthy children.

Comments:

Row 32-33: should be “… containing gluten (i.e. wheat, rye, barley)”. Others like spelt (wheat) are just varieties of them.

Rows 67-70: the title should be Physical activity (not Energy expenditure; you did not study it).

Was the information on physical activity collected by a questionnaire or by an interview? Self-reported?

Row: 79 Do you think the participants were able to explain the sugar and fat contents of foods they ate? Or was this inquired only for specific foods like milk, yoghurt where label about the content was available.

Row 81: the most critical point of this study is what type of food composition data was available. Main conclusions are based on the results calculated using food composition info. It would be good to get some more information about the food composition database used. You mention that you used food composition database, an Excel spreadsheet that was specifically developed for this study. How many foods did you have in the database? Did you take into consideration e.g. the variation of fiber content in various cereal products (whole grain breads, “white” bread, cakes, muffins, shortbreads, biscuits, donuts, crackers, pies, pasta of various types, pizza crust of various types, porridges, muesli, snacks of various types)? What was the source of the nutritional composition of foods? Does your country have their own national food composition database where to retrieve the information? How did you estimate the weight of the foods eaten (f.ex. 2 dl of cooked rice pasta, or a slice of oatmeal bread) in order to match the weight which was used to indicate the energy and macronutrient intake (often expressed per 100g of food). For example the nutrient values per 100 g of potato is very different depending how it has been prepared (the yield varies): boiled, roasted, baked, mashed, pan fried, deep fried, dried. Maybe you had the portion size photos and nutritional info covering most of the foods. It would be good to have more information about assessing the macronutrient values.

When estimating the frequency of food intake how did you record e.g. a vegetable omelet where the weight of eggs and vegetables are about 1:1? Did you count it both as eggs and as vegetables? What about eggs in a cake? Would it go to both egg and cereal category?

Row 88: do you mean you included tubers to cereals? It would not be correct.

Row: because KIDMED index is probably a new concept to many readers it would be good to have a little bit more explanation about the 16-point questionnaire, maybe as an attachment (online material).

Figure 1  needs still work. Numbers don’t match. A line missing. Some of the text is hidden.

Table 1. Please make decision which expression you want to use e.g. first to third quartiles, 1;3 quartiles or IQR and then use it systematically.

Row 147: Please use “fats” instead of “lipids”.

Row 151: Prevention of what?

Table 2: IC 95% ? - please correct.

Do you mean “simple sugars” or “added sugars”? If it refers to all sugars use “total sugar”. Also total proteins, total fats, total carbohydrates, total fiber.

Table 3: Is this about daily food consumption frequency? Make it clearer. The 3-day food record is not very good in estimating the consumption frequency of foods, e.g. foods that are not eaten daily (e.g. legumes, fish, eggs). Usually food consumption frequencies are estimated using food frequency questionnaire that gives better picture about the habitual food intake (frequency), although it overestimates the daily intake. However, it depends on the local food culture. P-values have both commas and periods. What are “minor cereals”?

Figure 2. shows that fiber intake is fairly high from gluten-free products (about 60% of all the fiber intake) and thus does not support the conclusion. If you prepare similar figure for healthy children showing their fiber intake from all commercial regular cereal products it may contrast the status better.

Maybe the best way would be to present this figure for CD children and for health children separately but taking into consideration all the cereals in both groups of children. It should show that the fiber intake from cereals (which are gluten free for CD children if the adherence to GF diet is 100%) is lower among CD children compared to the control children (who have mostly regular cereals).

One thing that you have not reported is the adherence to gluten-free diet among CD children. You can see it from the food records if they reported any product containing wheat, rye or barley. If they honestly report all their food consumption I don’t think the adherence of all the children is 100% (based on the experience from other studies in children). This would be an important information to report.

Rows 215-236: Could this be summarized instead of mentioning every study and detail separately?

Row 236: Reference for 3-day food record being the “best-practice” method?

Row 242: Oats are considered gluten-free in this study based on the introduction. You are not commenting them or rice which overall are much more important cereals in GF diet than pseudocereals.

Thank you!

Reviewer 2 Report

The unbalance of gluten-free diet is a mattern of concern for health professionals, and these kind of studies are of great interest in order to identify weaknesses of nutritional approaches of this collective. The paper is clearly written.

Major comments:

Line 81. How did you measure the macronutrient content of food? You specified that you used an ad-hoc excel spreadsheet, but you did not specify which database you use. Please provide a reference for the information source. Section 3.5. How did you calculate the contribution of commercial gluten-free products to the diet? Considering label information? Please specify in the text. Moreover, I recommend to add some comment regarding the necessity of enhancing nutritional quality of these products at the end of the discussion.

Minor comments:

Line 76: You mentioned that you used a specifically developed diary for CD patients. What is the difference between your diary and a standard one? Please give a reference for this diary if it is available. Figure 1. Please check the format, some text is hidden. I suppose that the information is hidden, but the number of children in the refusal group did not sum 25. Table 2. Please correct mistakes: Control group lacks the l º symbol is not appropriate for quartiles Energy: line below median and quartiles and above IC is not appropriate Table 3. Control group lacks the l Line 258. Please provide references for the fiber content of different cereals. Conclusions: it is important to highlight that celiac children need dietary counseling as control children do, as important unbalances are identified in the study for them.
